# Neonatal Exposure to Valproate Induces Long-Term Alterations in Steroid Hormone Levels in the Brain Cortex of Prepubertal Rats

**DOI:** 10.3390/ijms24076681

**Published:** 2023-04-03

**Authors:** Soon-Ae Kim, Eun-Hye Jang, Jangjae Lee, Sung-Hee Cho

**Affiliations:** 1Department of Pharmacology, School of Medicine, Eulji University, Daejeon 34824, Republic of Korea; dmter12@gmail.com; 2Chemical Analysis Center, Korea Research Institute of Chemical Technology (KRICT), Daejeon 34114, Republic of Korea; jjlee714@krict.re.kr; 3Department of Chemistry, Korea University, Seoul 02841, Republic of Korea

**Keywords:** valproic acid, steroid hormone, neonatal, sex-specific difference, autism spectrum disorder, neurosteroid

## Abstract

Valproic acid (VPA) is a known drug for treating epilepsy and mood disorders; however, it is not recommended for pregnant women because of its possible teratogenicity. VPA affects neurotransmission and gene expression through epigenetic mechanisms by acting as a histone deacetylase inhibitor and has been used to establish animal models of autism spectrum disorder (ASD). However, studies on the long-term effects of early exposure to VPA on glucocorticoid and neurosteroid synthesis in the brain are lacking. Therefore, this study aimed to investigate the long-term changes in metabolic alterations and gene expression regulation according to sex, using metabolic steroid profiling data from cerebral cortex samples of rats four weeks after VPA exposure (400 mg/kg). In neonatal VPA-exposed models, estradiol levels decreased, and cytochrome P450 19A1 gene (*Cyp19a1*) expression was reduced in the prepubertal male cortex. Progesterone and allopregnanolone levels decreased, and 3β-hydroxysteroid dehydrogenase 1 gene (*Hsd3b1*) expression was also downregulated in the prepubertal female cortex. Furthermore, cortisol levels increased, and mRNA expression of the nuclear receptor subfamily 3 group C member 1 gene (*Nr3c1*) was downregulated in the cortices of both sexes. Unlike the neonatal VPA-exposed models, although a decrease in progestin and estradiol levels was observed in females and males, respectively, no differences were observed in cortisol levels in the cortex tissues of 8-week-old adult rats administered VPA for four weeks. These results indicate that early environmental chemical exposure induces long-term neurosteroid metabolic effects in the brain, with differences according to sex.

## 1. Introduction

Valproic acid (VPA) is a known drug for treating epilepsy and mood disorders; however, it is not recommended for pregnant women because of its possible teratogenicity. VPA has been associated with behavioral teratogenicity, including autism spectrum disorder (ASD) [1]. ASD is a representative neurodevelopmental disorder whose etiology includes various genetic and environmental factors. Various genetic animal models have been used to elucidate its pathophysiology, and there continues to be studies on the effects of maternal stress and environmental factors in perinatal pregnancy on the onset of neurodevelopmental disorders such as ASD. Given that VPA-exposed rodents express an ASD-like superficial behavioral phenotype, VPA-exposed rodent models were suggested as valid representations of human ASD models in a meta-analysis [2].

Moreover, VPA acts as a histone deacetylase (HDAC) inhibitor, which is hypothesized to induce long-term effects through epigenetic mechanisms such as histone acetylation and alter the expression of transcription factors, leading to alterations in gene expression related to the cell cycle or brain neuronal differentiation in neurodevelopment [3]. Early neonatal VPA exposure from postnatal day (PND) 2 to 4 has been reported to stimulate the proliferation of glial precursors during cortical gliogenesis and affect hyperactivity and social interaction behaviors at PND 21 or 22 [4,5].

Several studies have also reported VPA as a mitochondrial toxicant, which affects steroid synthesis through a complex mechanism that mainly involves cholesterol access to the inner mitochondrial membrane [6]. The VPA-mediated increase in basal steroidogenesis could be associated with increased basal cortisolemia, which has been described in VPA-treated patients [7]. Furthermore, acute VPA exposure increases cortisol levels in the brain cortices of 4-week-old prepubertal mice [8]. As an environmental factor, perinatal stress has been suggested as a cause of neurodevelopmental disorders, and VPA may induce hypercortisolemia at the time of exposure. However, studies on the long-term effects of early VPA exposure on glucocorticoid and neurosteroid synthesis in the brain are lacking.

Therefore, this study aimed to investigate the long-term changes in steroid hormone and related gene expression levels using metabolic steroid profiling data from cerebral cortex samples from rats euthanized four weeks after VPA exposure (PND 2 to 4, PND 28).

## 2. Results

### 2.1. Increased Cortisol Levels Were Observed in the Cerebral Cortices of Neonatal VPA-Exposed 4-Week-Old Rats

To investigate the alterations in metabolic steroids in male and female brains following early neonatal VPA exposure, LC-MS/MS analysis was performed (Appendix A). In the 4-week-old male rats, the early VPA-exposed group exhibited increased cortisol levels (controls: 5.9 ± 1.8 ng/g, VPA: 10.2 ± 3.9 ng/g; *p* < 0.002, Figure 1A) and significantly decreased 17β-estradiol levels (controls: 0.20 ± 0.08 ng/g, VPA: 0.12 ± 0.03 ng/g; *p* < 0.004, Figure 1B). In the cortices of the early VPA-exposed female rats, the cortisol (controls: 6.1 ± 1.7 ng/g, VPA: 9.0 ± 2.8 ng/g; *p* < 0.03, Figure 1C) and cortisone (controls: 2.3 ± 0.8 ng/g, VPA: 3.9 ± 1.3 ng/g; *p* < 0.01, Figure 1D) levels were significantly increased, whereas those of progesterone (controls: 10.0 ± 3.2 ng/g, VPA: 6.3 ± 1.8 ng/g; *p* < 0.007, Figure 1E) and allopregnanolone (controls: 11.6 ± 2.3 ng/g, VPA: 8.1 ± 2.1 ng/g; *p* < 0.005, Figure 1F) were significantly decreased, compared to those in the controls. 

### 2.2. Decreased Expression of Cyp19a1 and Hsd3b1 According to Sex Was Observed in the Neonatal VPA-Exposed Cortex

An experiment was conducted to determine whether the changes in steroid hormone levels were due to alterations in the expression of metabolizing enzymes, such as aromatase (an enzyme that is encoded by the cytochrome P450 19A1 gene (*Cyp19a1*) and catalyzes the conversion of androgen to estrogen) and 3β-hydroxysteroid dehydrogenase (an enzyme that is encoded by *Hsd3b1* and is involved in progesterone synthesis). The mRNA expression levels of *Cyp19a1* and *Hsd3b1* were measured using real-time qPCR with neonatally exposed 4-week-old rat cortex samples. There were no significant difference in the mRNA expression of *Cyp19a1* in female cortices and *Hsd3b1* in male cortices between the control and the VPA-exposed group (Appendix A). However, decreased *Cyp19a1* mRNA levels were observed only in the VPA-exposed male cortices (Figure 2A; *p* < 0.001), and decreased *Hsd3b1* mRNA levels were observed only in the VPA-exposed female cortices (Figure 2B; *p* = 0.003). Furthermore, decreased CYP19A1 protein levels in the male cortices (Figure 2C; *p* = 0.015) and decreased HSD3B1 level in the female cortices (Figure 2D; *p* = 0.027) were confirmed using Western blot analysis.

### 2.3. Decreased Expression of Glucocorticoid Receptors Was Observed Only in the Neonatal VPA-Exposed Male Cortex

Experiments were conducted to determine the mRNA expression levels of GR (encoded by the nuclear receptor subfamily 3 group C member 1 gene [*Nr3c1*]), a major intracellular receptor for glucocorticoid steroid hormone, using real-time qPCR and Western blotting. Decreased *Nr3c1* mRNA levels were observed in both the male and female cortices (Figure 3; male, *p* = 0.034; female, *p* = 0.047). Decreased NR3C1 protein levels were confirmed only in the neonatal VPA-exposed male cortex using Western blot analysis (Figure 3; *p* = 0.035).

### 2.4. No Differences Were Observed in the Cortisol Level and Nr3c1 mRNA Expression in the Cerebral Cortices of Prepubertal VPA-Exposed 8-Week-Old Rats

Although the cortisol levels were significantly increased without sex-specific differences in the cortices of neonatal VPA-exposed 4-week-old rats compared with those in the controls, no differences were observed in cortisol levels in the cerebral cortices of prepubertal VPA-exposed 8-week-old rats (Figure 4A,C). In addition, the mRNA expression of *Nr3c1* remained unchanged, even in the male cortex (Appendix A). Significant increases in *Hsd3b1* mRNA levels were observed in the cerebral cortices of prepubertal VPA-exposed 8-week-old rats (Figure 5A,B; male, *p* = 0.007; female, *p* < 0.001).

However, the mRNA levels of *Cyp19a1* were not significantly different in the 8-week-old cortices (Figure 5C,D), although the levels of 17β-estradiol (controls: 0.26 ± 0.10 ng/g, VPA: 0.14 ± 0.07 ng/g; *p* < 0.01, Figure 4B) in the 8-week-old male cortex and the progesterone (controls: 12.1 ± 3.4 ng/g, VPA: 7.5 ± 2.7 ng/g; *p* < 0.01, Figure 4E) and allopregnanolone (controls: 12.4 ± 3.1 ng/g, VPA: 7.6 ± 1.4 ng/g; *p* < 0.001, Figure 4F) in the 8-week-old female cortex were still significantly decreased compared to those in the controls, similar to that observed in the neonatal VPA-exposed models (Appendix A).

## 3. Discussion

Steroid hormones control many physiological processes, such as stress responses, and are mainly produced by the endocrine glands [9]. Moreover, steroids produced within the nervous system are termed “neurosteroids” [10]. During neurosteroid synthesis, cholesterol metabolized to progesterone by 3β-HSD can be further metabolized to androgens (androstenedione and testosterone) and transformed into estrogens through aromatization [11], Figure 6. Estrogens potentiate glutamate responses primarily by bolstering N-methyl-d-aspartate (NMDA) receptor activity, affecting GABAergic mechanisms and altering brain morphology by increasing dendritic spine density [11]. Progesterone and allopregnanolone, endogenous 5α-reductase metabolites, are potent modulators of the γ-aminobutyric acid A (GABA_A_) receptor, and several studies have reported that preterm birth causes major deficits in GABA_A_ receptor neurosteroid signaling processes in the developing brain, leading to reduced exposure to the normal neurosteroid environment during late pregnancy [12,13].

Extensive evidence from animal studies has established an association between perinatal glucocorticoid exposure and subsequent psychopathologies. In addition, several human studies have demonstrated that postnatal glucocorticoid exposure leads to altered hypothalamic pituitary adrenal (HPA) axis activity and behavioral problems in childhood [14]. Glucocorticosteroids such as cortisol are modulators of CYP enzymes, and the altered activity of these enzymes can cause changes in neurosteroid synthesis [15]. Moreover, steroid hormones, which are ligands for a family of nuclear transcription factors, have been suggested to recruit numerous coactivators and repressors, including HDACs, to broad brain regions [16]. Therefore, changes in the neurosteroid levels in the brain may induce long-term secondary epigenetic alterations. Studies on guinea pigs demonstrated that prenatal GR expression was observed from gestational day 40 to PND 7. Additionally, recent studies have provided information on the specific human developing brain cell types that are positive for *Nr3c1* transcripts using single-cell RNA sequencing (sc-RNA-seq) of human fetal samples and cerebral organoids. GR exhibits cell type-specific expression in the nervous system and is expressed in various neuronal cells, including mature but not immature neurons [17]. Molecular mechanisms hypothesized to underlie the programming effects of early life stress and glucocorticoids include epigenetic changes in target chromatin, affecting the tissue-specific expression of the intracellular GR [18]. 

Hamden et al. (2021) reported that there was no statistical significance in corticosterone levels detected in three different brain regions (hippocampus, cerebral cortex and hypothalamus) at PND4,21 and 90 mice. It is expected that the results from the cortex tissue sample could represent overall brain steroid profiling [19]. Recently, Hamden et al. (2022) also suggested that early life stress by isoflurane exposure in the neonatal period (postnatal day 1 to 12) induced a neurosteroid production change in the brain and the effects on brain and behavior [20]. In this study, increased cortisol and decreased allopregnanolone levels were observed in the cortices of neonatal VPA-exposed rats. Lee et al. (2016) reported the stimulation of glial precursor proliferation and increased numbers of astrocytes with behavioral changes in the same animal model [4]. Moreover, neurosteroids stimulate oligodendrocyte maturation and myelination [12]. Repeated glucocorticoid exposure has been reported to suppress the expression levels of 5α-reductase and allopregnanolone in the fetus, resulting in reduced myelination. Preterm birth induces an abrupt loss of protective effects of allopregnanolone, with a marked decline in allopregnanolone concentrations in the preterm neonatal brain compared to that in the fetal brain. Additionally, preterm infants have been confirmed to have altered cerebral myelination, with reductions in white matter volumes in the frontal cortex, hippocampus, and cerebellum, as evident by magnetic resonance imaging (MRI) [21]. Furthermore, maternal cortisol levels during pregnancy have been reported to be related to newborn amygdala architecture and connectivity in a sexually dimorphic manner [22].

In this study, neonatal VPA exposure decreased aromatase expression and estradiol levels only in the prepubertal male cortex. Estrogen activity is not limited to women and has been consistently reported in men. A decreased 17β-estradiol level may affect the expression of genes associated with synaptic activity, myelination, and neurotransmission, and 17β-estradiol may regulate signaling pathways involved in corticogenesis, with induction of neuronal stem cell proliferation as a key molecular player. During early development, 17β-estradiol decreases GABAergic signaling and induces excitatory synapse formation [23,24]. It has also been suggested that 17β-estradiol plays a pivotal role in sex-typical and socio-aggressive behaviors and the development of sexually dimorphic regions, such as the cerebral cortex, during early brain development [25].

Studies have also suggested that sex-related differences in the enzymatic activities in steroid hormone metabolism are induced by epigenetic mechanisms involving VPA as an HDAC inhibitor. Although predicting the direction of changes in gene expression is difficult, it has been suggested that HDAC inhibitors affect the expression of numerous genes in the hippocampus [26]. For example, VPA induces long-term alterations in gene expression, blocking the masculinizing testosterone actions in specific brain regions and indicating sex-related differences in cell number [27]. Furthermore, VPA-induced ASD-like behavioral changes can be epigenetically transmitted to the third generation [28]. 

In summary, sex-specific differences were observed in the neonatal VPA-induced alterations of neurosteroid levels in the brain cortices of prepubertal rats. With the exception of cortisol, these sex-specific differences were observed in the brain cortices of adult rats exposed to VPA at a prepubertal age. Further studies with animal models of various ages and brain regions are required to better understand the biological, toxicological, and behavioral effects and mechanisms underlying the changes in steroid hormone expression. These results may provide a better understanding of the VPA-induced sex-specific differences in the metabolic alterations of steroids.

## 4. Materials and Methods

### 4.1. Animals and Treatments

This study was approved by the Institutional Animal Care and Use Committee of Eulji University (EUIACUC 20-05). Eight-week-pregnant and three-week-old Sprague-Dawley (SD) rats (male: *n* = 8, female: *n* = 8) were purchased from Samtako Inc. (Gyeonggi-do, South Korea) and maintained on a standard regular 12-h light–dark cycle at ambient temperature (22 °C ± 2 °C) and humidity (50% ± 10%), with food and water available ad lib. Pregnant rats were randomly divided into two groups and checked every morning; the day of delivery of newborn pups was considered PND 0. Litter size was usually between 7 and 15. After delivery, each female was allowed to individually raise their litter until PND 21. The VPA treatment procedure was conducted according to a method described by Lee et al. [5]. Briefly, pups born to the same mothers were subcutaneously administered treatment (300 mg/kg VPA in saline solution or normal saline) twice daily on PND 2 and 3 and once on PND 4. Next, the 3-week-old rats were randomly divided into 4 groups (each group: *n* = 8) to investigate VPA-induced metabolic alterations according to sex. Each experimental group was administered VPA (300 mg/kg in saline solution) subcutaneously, and each control group (male: *n* = 8, female: *n* = 8) was administered saline solution subcutaneously five times for three days. Four weeks after treatment, the rats were euthanized under isoflurane anesthesia (Hana Pharm, Seoul, South Korea) and decapitated to obtain brain samples. The cerebral cortex was isolated from each brain, and cortex samples were stored at −80 °C until analysis.

### 4.2. Analysis of Steroids in Cerebral Cortex Samples Using LC-MS/MS

For steroid analysis, the cerebral cortex samples were homogenized in a 1:9 (*w/v*) MeOH/acetic acid (99:1 *v/v*) mixture. After incubation for extraction at 4 °C overnight, the samples were centrifuged at 12,000 rpm for 5 min, and the pellet was extracted twice with 1 mL of a 99:1 (*v/v*) MeOH/acetic acid mixture. The organic phases were combined and dried under a nitrogen stream, after which the samples were resuspended in 1 mL of a 10:90 (*v/v*) MeOH/water mixture and extracted using Oasis PRiME HLB solid-phase extraction (SPE) cartridges with a peristaltic pump. The samples were loaded onto a cartridge, washed with 1 mL of water, and eluted twice with 1 mL of MeOH. The combined methanol eluates were evaporated under a nitrogen stream and dissolved in 100 µL of MeOH. Finally, 5 µL of the solution was injected into the liquid chromatography-tandem mass spectrometry (LC-MS/MS) system. The LC-MS/MS analysis was performed using a Waters^®^ Acquity UPLC I-Class system (Waters Corporation, Milford, MA, USA) that was interfaced with a Waters^®^ Xevo TQ-S micro tandem mass spectrometer (Waters Corporation) with electrospray ionization (ESI) source. Chromatographic separation was achieved with a Waters^®^ Acquity UPLC BEH C18 octadecylsilane column (2.1 mm × 100 mm, 1.7 μm). The LC conditions for the separation were as follows: mobile phase A was 0.2 mM ammonium fluoride in H_2_O, and mobile phase B was 0.2 mM ammonium fluoride in methanol. The gradient program (*v/v*) had a flow rate of 400 μL/min and was started with 30% B (*v/v*), held at 30% B (*v/v*) for 5 min, increased to 50% B (*v/v*) at 15 min, held at 50% B (*v/v*) for 5 min, increased to 55% B (*v/v*) at 23 min, increased to 80% B (*v/v*) at 27 min, and then held at 80% B (*v/v*) for 3 min. The column was re-equilibrated for 3 min with 30% B (*v/v*). The temperature of the column was maintained at 40 °C. Separated steroid hormones were monitored by positive (for androgens, corticoids, and progestins) and negative (for estrogens) electrospray ionization (ESI) tandem mass spectrometry (MS/MS). The source and operating parameters were optimized as follows: capillary voltage of 3 kV (ESI+) and 4 kV (ESI-); cone gas flow of 30 L/hr; desolvation temperature of 450 °C; and desolvation gas flow of 800 L/h. The quantitative analysis was performed in multiple reaction monitoring (MRM) mode, and peak identifications were achieved by comparing the retention times and matching the MS/MS ions. Data acquisition was performed with MassLynx software (V4.1, Waters Corporation). The quantification of steroids was performed according to a previous study [8], and the calibration data are shown in Appendix A.

### 4.3. RNA Extraction and Quantitative Real-Time qPCR Analysis

Total RNA was isolated from the tissues using an RNeasy Mini Kit (Qiagen, Hilden, Germany). Complementary DNA (cDNA) was synthesized from 100 ng of total RNA using an iScriptTM cDNA synthesis kit (Bio-Rad, Hercules, CA, USA), according to the manufacturer’s instructions. Complementary DNA was mixed with SYBR Green super mix (Bio-Rad, Hercules, CA, USA) and primers and amplified using a CFX96TM Real-Time System (Bio-Rad, Hercules, CA, USA). The sequences of the primers used for amplifying target genes are included in Appendix A. The following qPCR conditions were used: 95 °C for 3 min, 40 cycles of 95 °C for 15 s, and annealing temperature of 1 min. Raw data were analyzed using the delta–delta cycle threshold (2^−ΔΔCT^) method. The reference gene validation was performed with RefFinder program using housekeeping genes (https://blooge.cn/RefFinder/?type=reference, accessed on 10 February 2023). Glyceraldehyde-3-phosphate dehydrogenase gene (*Gapdh*) was used as a normalizer for all CT values and was calculated as the fold change relative to the CT value of the control.

### 4.4. Western Blotting

Tissue proteins were extracted using radioimmunoprecipitation assay (RIPA) buffer (ATTO, Tokyo, Japan) with proteinase and phosphatase inhibitors (ATTO, Tokyo, Japan). Protein concentration was measured using a bicinchoninic acid (BCA) assay (Thermo Scientific, Waltham, MA, USA). The proteins (20 µg) were separated on 10% SDS-PAGE gels and transferred to nitrocellulose membranes (Pall, Port Washington, NY, USA). The membranes were blocked with 5% non-fat milk in Tris-buffered saline with Tween (TBST) buffer for 1 h at 4 °C and incubated with primary antibody diluted in TBST at 4 °C overnight. The membranes were rinsed with TBST buffer and incubated with horseradish peroxidase (HRP)-labeled secondary antibody diluted in TBSTfor 1 h. The antibodies are presented in Appendix A. Next, the membranes were incubated with West Femto Maximum Sensitivity Substrate (Thermo Scientific, Waltham, MA, USA). Protein expression was detected by exposure to X-ray film (Agfa, Mortsel, Belgium) and analyzed using ImageJ lab software (Bio-Rad, Hercules, CA, USA).

### 4.5. Statistical Analysis

Data manipulation was performed using Excel 2013 spreadsheets (Microsoft Corporation, Seattle, WA, USA) and GraphPad Prism 7 (GraphPad Software Inc., La Jolla, CA, USA). The quantitative results are expressed as mean ± standard deviation, and the groups were compared using unpaired two-tailed Student’s *t*-tests using the Statistical Package for the Social Sciences (SPSS) v20 (IBM, Armonk, NY, USA). *p*-values ˂ 0.05 were considered statistically significant.

## Figures and Tables

**Figure 1 ijms-24-06681-f001:**
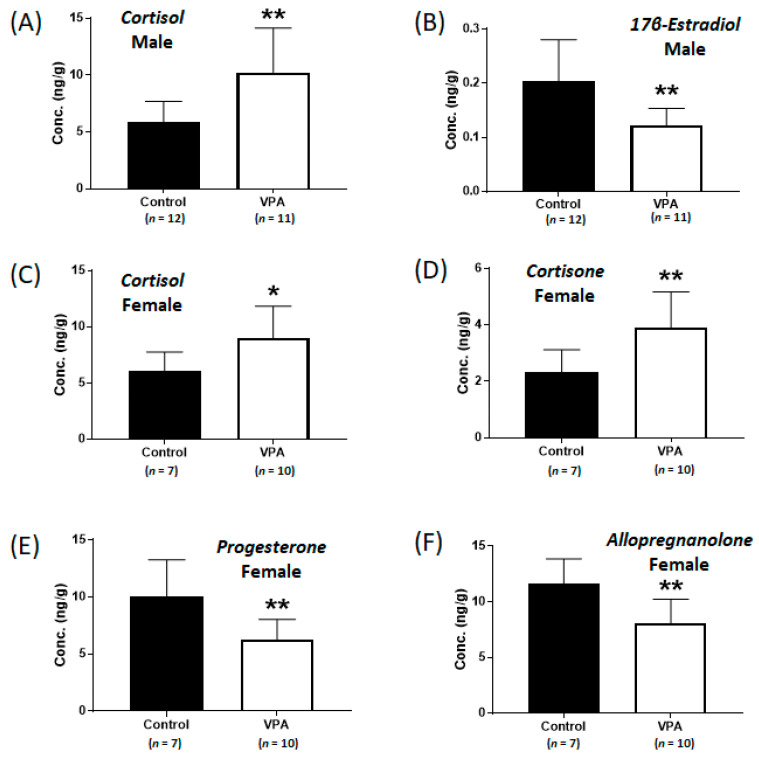
Concentrations of cortisol (**A**) and 17β-estradiol (**B**) in neonatal valproic acid (VPA)-exposed 4-week-old male rat cortex and concentrations of cortisol (**C**), cortisone (**D**), progesterone (**E**), and allopregnanolone (**F**) in neonatal VPA-exposed 4-week-old female rat cortex compared to those in the controls. Values represent the mean ± standard deviation (SD) of individual samples. The significance level between the control and the exposure groups is indicated by * *p* < 0.05, ** *p* < 0.01.

**Figure 2 ijms-24-06681-f002:**
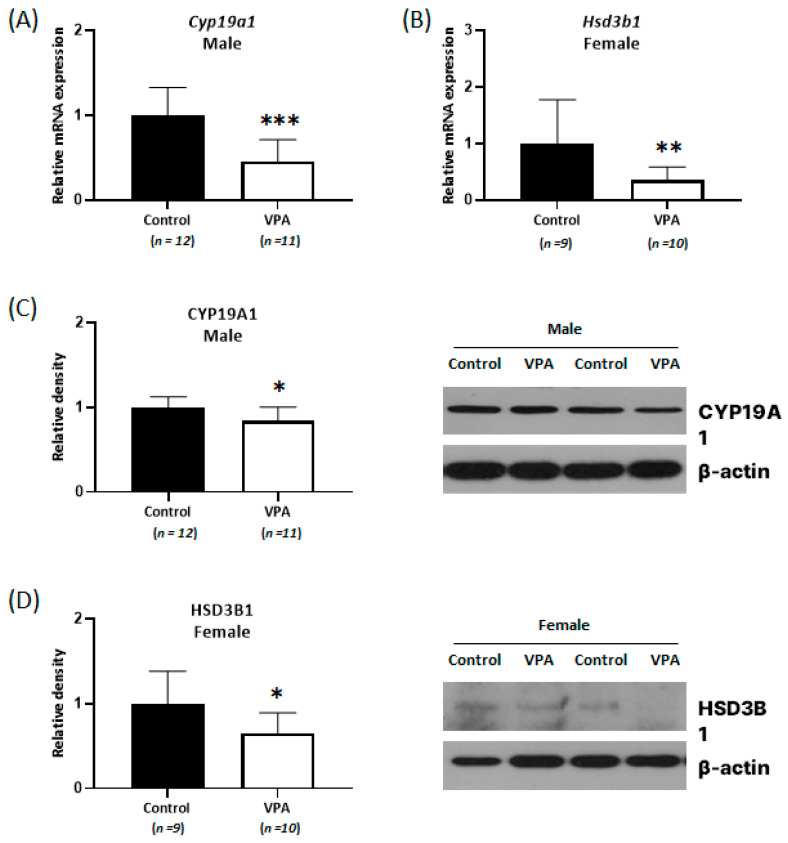
Relative messenger RNA (mRNA) and protein expression of *Cyp19a1* (**A**,**C**) and *Hsd3b1* (**B**,**D**) decreased with sex-related differences in neonatal VPA-exposed 4-week-old rat cortices. Relative mRNA expressions were measured by real-time qPCR. Representative Western blots (bottom). β-actin was used as the loading control to normalize the content of each sample. * *p* < 0.05, ** *p* < 0.01, *** *p* < 0.001.

**Figure 3 ijms-24-06681-f003:**
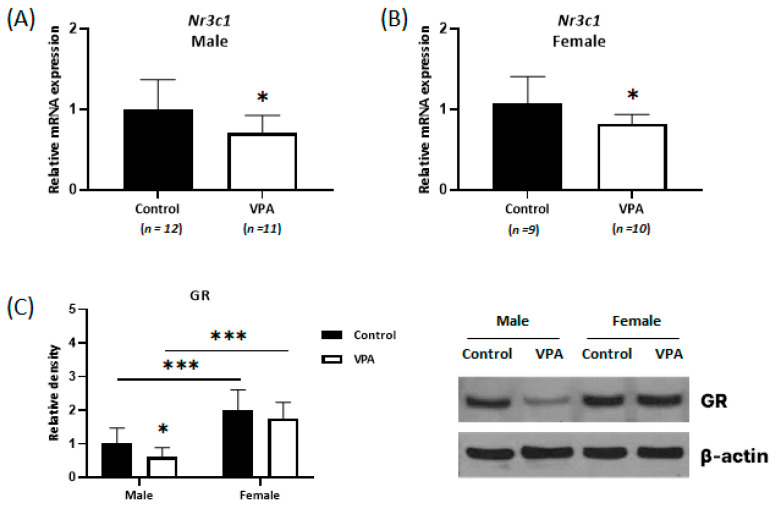
Decreased expression of glucocorticoid receptors (GRs) was observed only in the neonatal VPA-exposed male cortex. Relative expressions of *Nr3c1* mRNA measured by real-time qPCR in the neonatal VPA-exposed 4-week-old rat cortex (**A**,**B**). Representative Western blots (**C** Right). Quantification of GR protein levels in the cortex (**C** Left). β-actin was used as the loading control to normalize the content of each sample. * *p* < 0.05, *** *p* < 0.001.

**Figure 4 ijms-24-06681-f004:**
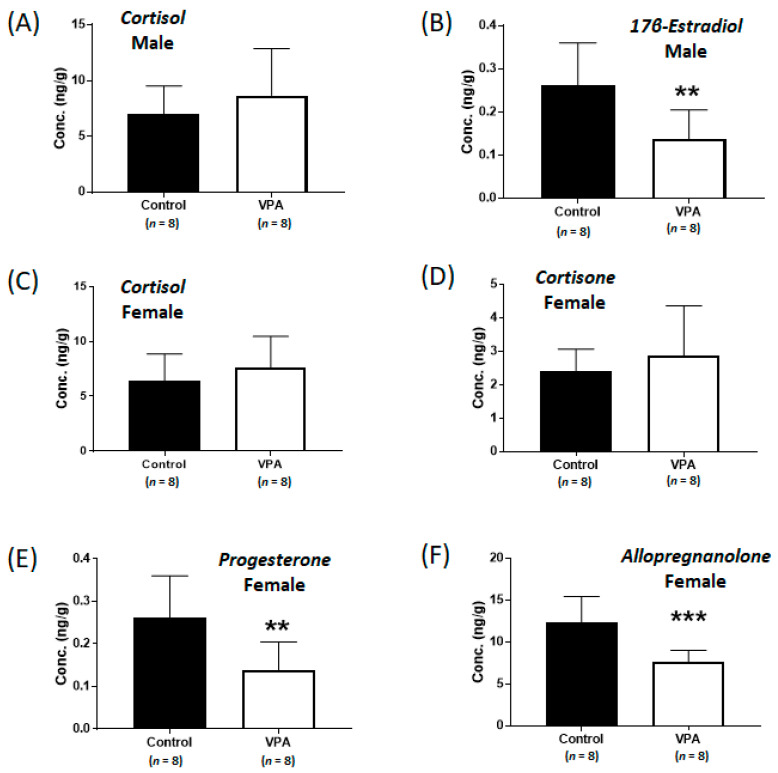
Concentrations of cortisol (**A**) and 17β-estradiol (**B**) in the cerebral cortices of prepubertal VPA-exposed 8-week-old male rats and concentrations of cortisol (**C**), cortisone (**D**), progesterone (**E**), and allopregnanolone (**F**) in the cerebral cortices of prepubertal VPA-exposed 8-week-old female rats compared to those in the controls. Values represent the mean ± standard deviation (SD) of individual samples. The significance level between the control and exposure groups is indicated by ** *p* < 0.01, *** *p* < 0.001.

**Figure 5 ijms-24-06681-f005:**
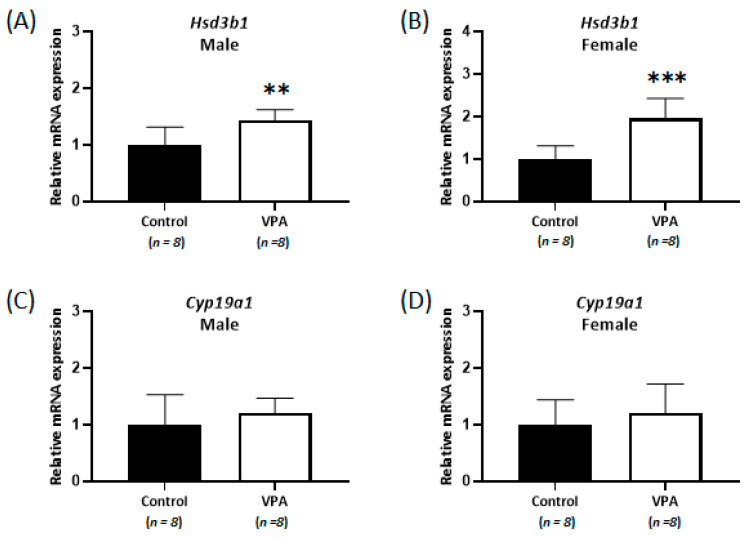
Relative mRNA expression of *Hsd3b1* (**A,B**) and *Cyp19a1* (**C,D**) measured by real-time qPCR in prepubertal VPA-exposed 8-week-old male and female rat cortices. No differences were observed in the cortisol level and *Nr3c1* mRNA expression in the cerebral cortices. ** *p* < 0.01, *** *p* < 0.001.

**Figure 6 ijms-24-06681-f006:**
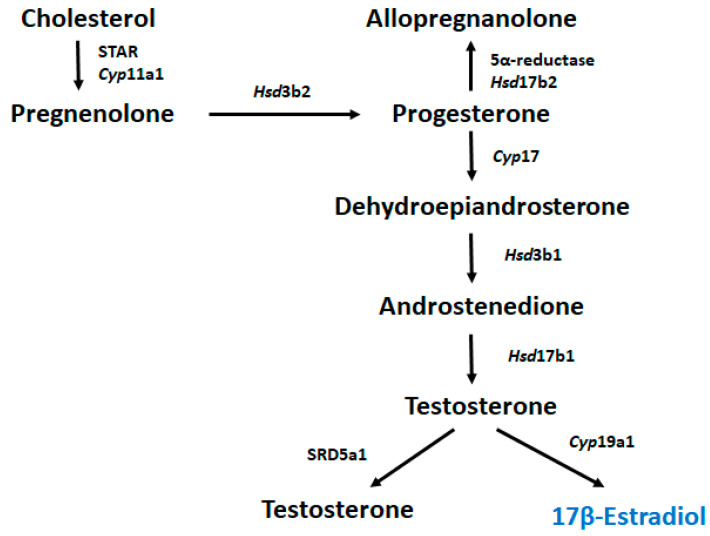
Metabolic pathway of 17β-estradiol biosynthesis.

## Data Availability

The authors declare that all required data have been presented in the manuscript. The datasets did not contain any software code needing to be archived.

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
