# Peer review of "Neonatal Exposure to Valproate Induces Long-Term Alterations in Steroid Hormone Levels in the Brain Cortex of Prepubertal Rats"

_ijms, 2023, doi:10.3390/ijms24076681_

Round 1

Reviewer 1 Report

The publication "Neonatal exposure to valproate induces long-term alterations in steroid hormone levels in the brain cortex of prepubertal rats" is written correctly and the research is done properly. However, I have a few questions:

1. Why was the expression of CYP19A1 in females and HSD3B1 in males not measured (Fig 2), while Fig 5 in another group shows the results?

2. Can the authors add a figure with the main pathways of 17β-estradiol synthesis? Such description is missing, it can be easily referred to.

3. Glucocorticosteroids such as cortisol are modulators of CYP enzymes [Danek PJ, Bromek E, Daniel WA. The Influence of Long-Term Treatment with Asenapine on Liver Cytochrome P450 Expression and Activity in the Rat. The Involvement of Different Mechanisms. Pharmaceuticals (Basel). 2021 Jun 29;14(7):629. doi: 10.3390/ph14070629. Altered activity of these enzymes can cause changes in neurosteroid synthesis. Please add a relevant paragraph.

4. Why was only one brain structure studied?

Reviewer 2 Report

1. How was the calibration performed in the quantification of steroid levels in LC-MS/MS measurements?

2. Recently, Hamden et al. 2021, and 2022 showed the effect of isoflurane on steroid levels in juvenile rats. Did the authors check the effect of anesthesia with isoflurane on the tested levels of steroids, e.g. by comparing the concentrations in the control/examined animal after decapitation without anesthesia?

3. Western blotting. What antibodies (company) and in what dilution was used? Please complete the details of all antibodies used.

4. Fig.2. Relative density - was there a standardization to beta-actin? If so, please mark it on the Y-axis or put the information in the caption under the figure. For blots, please mark which bands belong to males and females.

5. Statistical analysis. Could the student's t-test be used for all statistically analyzed parameters? Did they meet the criteria for this test - please specify. SEM or SD was marked on the steroid charts. There is a difference between the text and the supplementary material.

6. How the reference gene validation was performed - GAPDH selection? Please provide the sequence of primers used as a panel of reference genes for GAPDH selection.

7. Please replace PCR with qPCR in the temperature profile description.

Round 2

Reviewer 2 Report

1. How was the calibration performed in the quantification of steroid levels in LC-MS/MS measurements?

The supplementary material should include the results of the current calibration according to which the authors standardized the present results.

2. Recently, Hamden et al. 2021, and 2022 showed the effect of isoflurane on steroid levels in juvenile rats. Did the authors check the effect of anesthesia with isoflurane on the tested levels of steroids, e.g. by comparing the concentrations in the control/examined animal after decapitation without anesthesia?

I think that the information that isoflurane affects the level of measured steroids should be included in the discussion. Such information would be useful for other researchers to consider the possible effects of the anesthetic in their experiments.

4. Fig.2. Relative density - was there a standardization to beta-actin? If so, please mark it on the Y-axis or put the information in the caption under the figure. For blots, please mark which bands belong to males and females.

I guess it's still wrong because where are the bands of female CYP19A1 and bands of male HSD3B1?

5. Statistical analysis. Could the student's t-test be used for all statistically analyzed parameters? Did they meet the criteria for this test - please specify. SEM or SD was marked on the steroid charts. There is a difference between the text and the supplementary material.

The authors did not present test results confirming the possibility of using the Student's t-test in the case of analyzing their data.

6. How the reference gene validation was performed - GAPDH selection? Please provide the sequence of primers used as a panel of reference genes for GAPDH selection.

Please prepare and present the validation of the reference gene in one of the programs listed below and include the sequence of the panel of genes used for the validation of the reference gene.

RefFinder ( DOI: 10.1007/s11103-012-9885-2 )

Genorm ( DOI: 10.1186/gb-2002-3-7-research0034 )

NormFinder ( DOI: 10.1158/0008-5472.can-04-0496 )

BestKeeper ( DOI: 10.1023/b:bile.0000019559.84305.47 )

The comparative delta-Ct method ( DOI: 10.1186/1471-2199-7-33 )

Round 3

Reviewer 2 Report

In statistical analyses, there is no place for acceptance or non-compliance with the test criteria based only on the experimenter's impression. Whether the analyzed data set meets the criteria for parametric tests or not should be confirmed by the results of proper tests: normal distribution test and equality of variances test. If the results of these tests do not confirm the normal distribution and equality of variance, then non-parametric tests should be used. I assume that the authors did not perform such analyses because they did not present the results (p values) of these tests. The explanations provided by the authors are unsatisfactory. Therefore results of the statistical analyses could be not valid.
